# Frequency of Common and Uncommon *BRAF* Alterations among Colorectal and Non-Colorectal Gastrointestinal Malignancies

**DOI:** 10.3390/cancers16101823

**Published:** 2024-05-10

**Authors:** Amit Mahipal, Michael H. Storandt, Emily A. Teslow, Ellen Jaeger, Melissa C. Stoppler, Zhaohui Jin, Sakti Chakrabarti

**Affiliations:** 1Department of Medical Oncology, University Hospitals Seidman Cancer Center, Case Western Reserve University, Cleveland, OH 44106, USA; sakti.chakrabarti@uhhospitals.org; 2Department of Internal Medicine, Mayo Clinic, Rochester, MN 55905, USA; storandt.michael@mayo.edu; 3Tempus AI, Chicago, IL 60654, USA; emily.teslow@tempus.com (E.A.T.); ellen.jaeger@tempus.com (E.J.); melissa.stoppler@tempus.com (M.C.S.); 4Department of Medical Oncology, Mayo Clinic, Rochester, MN 55905, USA; jin.zhaohui@mayo.edu

**Keywords:** BRAF, gastrointestinal cancer, colorectal cancer

## Abstract

**Simple Summary:**

The combination of dabrafenib plus trametinib received tumor-agnostic approval for patients with *BRAF V600E* mutations. *BRAF* alterations are well characterized in patients with colorectal cancer (CRC) but their role in other gastrointestinal (GI) cancers is not known. This study demonstrates that amongst non-CRC GI cancers, *BRAF* alterations are most commonly present in bile duct cancers and small intestinal cancers. *BRAF* amplifications and *BRAF* fusions are more commonly observed in non-CRC GI cancers than in CRC and can potentially be an attractive therapeutic target. The prognostic impact of *BRAF* alterations in non-CRC GI cancers needs to be further investigated.

**Abstract:**

Background: The predictive and prognostic role of *BRAF* alterations has been evaluated in colorectal cancer (CRC); however, *BRAF* alterations have not been fully characterized in non-CRC gastrointestinal (GI) malignancies. In the present study, we report the frequency and spectrum of *BRAF* alterations among patients with non-CRC GI malignancies. Methods: Patients with CRC and non-CRC GI malignancies who underwent somatic tumor profiling via a tissue-based or liquid-based assay were included in this study. Gain-of-function *BRAF* alterations were defined as pathogenic/likely pathogenic somatic short variants (SVs), copy number amplifications ≥8, or fusions (RNA or DNA). Results: Among 51,560 patients with somatic profiling, 40% had CRC and 60% had non-CRC GI malignancies. *BRAF* GOF alterations were seen more frequently in CRC (8.9%) compared to non-CRC GI malignancies (2.2%) (*p* < 0.001). Non-CRC GI malignancies with the highest prevalence of *BRAF* GOF alterations were bile duct cancers (4.1%) and small intestine cancers (4.0%). Among *BRAF* GOF alterations, class II (28% vs. 6.8%, *p* < 0.001) and class III (23% vs. 14%, *p* < 0.001) were more common in non-CRC GI malignancies. Among class II alterations, rates of *BRAF* amplifications (3.1% vs. 0.3%, *p* < 0.001) and *BRAF* fusions (12% vs. 2.2%, *p* < 0.001) were higher in non-CRC GI malignancies compared to CRC. Conclusions: Non-CRC GI malignancies demonstrate a distinct *BRAF* alteration profile compared to CRC, with a higher frequency of class II and III mutations, and more specifically, a higher incidence of *BRAF* fusions. Future studies should evaluate clinical implications for the management of non-CRC GI patients with *BRAF* alterations, especially BRAF fusions.

## 1. Introduction

The *B-raf proto-oncogene* (*BRAF*) encodes BRAF protein, which normally functions via the mitogen-activated protein kinase/extracellular signal-related kinase (MAPK/ERK) pathway downstream of growth-factor receptors to regulate cell growth and proliferation [1]. Activating *BRAF* alterations may result in constitutive activation of this pathway, and are implicated in oncogenesis, leading to an increase in tumor growth and metastasis [2]. Different classes of *BRAF* alterations have been identified, each with distinct clinical relevance. Class I mutations, the most common, include *BRAF* V600E alterations, which lead to the activation of *BRAF* monomers in a kinase-independent manner. Class II mutations result in the formation of activated *BRAF* dimers, also in an RAS-independent manner. Lastly, class III mutations result in kinase-inactivating heterodimers in an RAS-dependent manner [3,4,5,6].

Among gastrointestinal (GI) malignancies, *BRAF* mutations have been extensively studied in colorectal cancer (CRC), detected in about 8–12% of patients with metastatic disease [7]. *BRAF* mutations may be classified as class I (V600E), class II, and class III, with each class having a distinct mechanism and clinical phenotype. Class I mutations result in constitutive activation of BRAF monomers in an RAS-independent manner, class II mutations result in constitutively activate BRAF dimers in an RAS-independent manner, and class III mutations more avidly bind RAS and CRAF. The most common alteration, *BRAF* V600E, has been associated with right-sided primaries, and poorer survival [6]. Class II *BRAF* mutations have also been associated with a poorer prognosis, comparable to patients with class I mutations, while those with class III mutations have been found to have a higher frequency of left-sided disease, with a lower probability of nodal metastases [5].

*BRAF* alterations have been less commonly reported in non-CRC GI malignancies, and little is known regarding the frequency and spectrum of these mutations. With the continued development of targeted therapies, it is essential to recognize the incidence of *BRAF* mutation types, as class I and II alterations may be a valid target in a subset of patients. Recently, dabrafenib, a *BRAF* inhibitor, in combination with trametinib, an MEK inhibitor, received tumor-agnostic approval for advanced solid malignancies harboring *BRAF* V600E [8]. This further highlights the importance of recognizing the frequency of *BRAF* alteration among non-CRC GI malignancies, as well as the class of *BRAF* alteration.

In the present study, we utilized data to determine the frequency of *BRAF* mutations among non-CRC GI malignancies and the spectrum of these mutations, and determined if there are differences in the prevalence of these alterations in certain patient populations.

## 2. Methods

### 2.1. Cohort Selection

Patients with a primary diagnosis of a GI malignancy were identified from the Tempus database and filtered to those who received Tempus xT or xF testing prior to September 2023. GI malignancies were defined as primary tumor sites outlined in Table 1. Patients were considered to have CRC if their primary cancer site was in the colon, rectum, or rectosigmoid junction. The concept of a primary cancer for this study is defined as the cancer diagnosis for which the patient underwent sequencing. Patients were further filtered into those who had a gain-of-function (GOF) *BRAF* alteration detected on either assay. For patients with multiple sequencing reports, the most recent report on which a *BRAF* alteration was detected was chosen for analysis. GOF *BRAF* alterations were defined as a pathogenic/likely pathogenic somatic short variant, copy number amplification ≥8, or fusion (RNA or DNA).

### 2.2. Sequencing Assays

Next-generation sequencing was conducted using the Tempus xT and xF assays (Tempus AI, Chicago, IL, USA), as previously described [9,10,11,12,13,14]. Briefly, Tempus xT is a targeted, tumor/normal-matched DNA panel that detects single-nucleotide variants (SNVs), insertions and/or deletions (indels), and copy number variants (CNVs) in 648 genes, as well as chromosomal rearrangements in 22 genes with high sensitivity and specificity. Tempus xF is a targeted liquid biopsy DNA panel that identifies SNVs and indels in 105 genes, CNVs in 6 genes, and chromosomal rearrangements in 7 genes.

### 2.3. Tumor Mutation Burden

Tumor mutation burden (TMB) was calculated as previously reported [12]. Briefly, TMB was calculated by dividing the number of nonsynonymous variations by the size of the panel. All nonsilent somatic coding variations such as missense, indel, and stop-loss variants with coverage greater than ×100 and an allelic fraction greater than 5% are included in the count of nonsynonymous variations. Tumors were considered to have high TMB (TMB-H) if they had an adjusted TMB score of 10 mutations per megabase (mut/Mb) or more in tissue testing, or 15 mut/Mb or more in liquid testing.

### 2.4. Microsatellite Instability

The Tempus xT panel included probes for 239 microsatellites that are frequently unstable in tumors with mismatch repair (MMR) deficiencies. The microsatellite instability (MSI) classification algorithm used reads mapping to these frequently unstable regions to classify tumors into three categories: microsatellite instability-high (MSI-H), microsatellite stable (MSS), or microsatellite equivocal (MSE). This assay can be performed with paired tumor–normal samples or tumor-only samples. Both algorithms return the probability of the patient being MSI-H, which is then translated into an MSI status of MSI-H, MSS, or MSE. All loci with sufficient coverage were tested for instability, as measured by changes in the distribution of the number of repeat units in the tumor reads as compared to the normal reads using the Kolmogorov–Smirnov test. If *p* ≤ 0.05, the locus was considered unstable. The proportion of unstable loci was fed into a logistic regression classifier trained on tumor samples with clinically determined MSI statuses. The Tempus xF panel included probes for 227 microsatellites that are frequently unstable in tumors with MMR deficiencies. To detect MSI, the relative frequency and distribution are determined for any read containing repetitive sequences. To predict the probability of an unstable locus, a k-nearest neighbors model (with k = 100) is used along with normalized percent lower, mean lower, and mean log-likelihood metrics. The percentage of unstable loci is calculated from the probabilities of each sample, with >50% unstable loci considered MSI-high [10,11,13].

### 2.5. Statistical Analysis

Demographic and clinical characteristics are described as percentages or median with range and interquartile range and compared by chi-squared/Fisher’s exact tests or Wilcoxon rank sum tests, as applicable. The prevalence of MSI and TMB-H are described as percentages and compared using chi-squared tests. *BRAF* alteration types, class types, amino acid effects, and prevalence of co-mutations are described and compared similarly, with a false discovery rate correction for multiple testing. Analyses are two-sided with statistical significance evaluated at the 0.05 alpha level.

## 3. Results

### 3.1. Cohort Characteristics

Among 51,560 sequenced patients with a history of a GI malignancy, 40% were diagnosed with CRC (*n* = 20,656) and 60% were diagnosed with a non-CRC GI malignancy (*n* = 30,904). Among the *BRAF*-altered cohort, 73% underwent a tissue-based assay and 27% underwent a liquid-based (blood) assay. The most common non-CRC GI malignancies included cancers of the pancreas (14,300; 28%), bile duct (4923; 9.5%), esophagus (3.316; 6.4%), and stomach (3029; 5.9%). In our cohort, 17% of the patients had another prior malignancy (*n* = 432) with the most common being breast cancer (3.2%), prostate cancer (2.1%), hematologic malignancy (1.9%), and skin cancer (0.8%). The rates of multiple primary cancers were similar in the two groups.

*BRAF* GOF-altered patients diagnosed with CRC were compared to non-CRC GI malignancies. CRC patients were slightly older than those with non-CRC GI malignancies (median age of 67 vs. 66 years; *p* = 0.029). *BRAF*-altered CRC was more commonly found in females (57% vs. 45% female; *p* < 0.001) while *BRAF*-altered non-CRC GI malignancies were more often observed in males. Among *BRAF*-altered CRC patients, 85% were white, whereas only 77% of those with a *BRAF*-altered non-CRC GI malignancy were white. Notably, significant differences were observed across races between CRC and non-CRC GI malignancies (*p* < 0.001), with 11% of those with a non-CRC GI malignancy being black or African American, versus only 6.4% of those with *BRAF*-altered CRC. Sixty-nine percent with CRC had metastases prior to testing, while fifty-nine percent with a non-CRC GI malignancy had metastases prior to testing (*p* < 0.001). Characteristics of the *BRAF*-altered patients based on primary tumor location are described in Table 1.

### 3.2. Incidence of BRAF Gain-of-Function Alteration by Primary Tumor Location

*BRAF* GOF alterations were more frequently seen in patients with CRC, with higher prevalence compared to non-CRC GI malignancies overall (8.9% vs. 2.2%, *p* < 0.001). Non-CRC GI malignancies with the highest incidence of *BRAF* GOF alterations included bile duct cancers (4.1%) and small intestine cancers (4.0%). The frequency of *BRAF* GOF alteration by primary tumor location is described in Table 2 and Appendix A.

### 3.3. Characterization of BRAF Gain-of-Function Alterations

Among all patients with a *BRAF* GOF alteration, 94% were the result of a short variant (SV). SVs were more common among patients with CRC (98%) compared to patients with a non-CRC GI malignancy (85%) (*p* < 0.001). The most common SV mutations were in exon 15 and exon 11 with rates of 90% and 6.4%, respectively, in patients with CRC. The corresponding rates for patients with non-CRC GI cancers were 58% and 15%, respectively. Among patients with a *BRAF*-altered non-CRC GI malignancy, the prevalence of copy number amplification (3.1% vs. 0.3%, *p* < 0.001) and fusions (12% vs. 2.2%, *p* < 0.001) were higher when compared to those with CRC (Figure 1, Appendix A). Interestingly, there was also a small subset of patients with >1 BRAF alteration (SV, CNA, and/or fusion) detected, including four patients who harbored both a pathogenic BRAF fusion in addition to a V600E alteration. Figure 2 characterizes the type of *BRAF* alteration based on the primary location for non-CRC GI malignancies. Appendix A provides a full list of fusions detected.

When evaluating alteration class amongst *BRAF* GOF cancers, class I mutations were more common in *BRAF*-altered CRC compared to non-CRC GI malignancies (75% vs. 28%, *p* < 0.001). However, class II (28% vs. 6.8%, *p* < 0.001) and class III (23% vs. 14%, *p* < 0.001) were more commonly seen in *BRAF*-altered non-CRC GI malignancies compared to *BRAF*-altered CRC. Specific *BRAF* mutations with a prevalence of ≥1% are reported in Table 3.

### 3.4. MSI and TMB among BRAF-Altered Tumors

We then characterized TMB and MSI status in *BRAF*-altered malignancies. TMB-H was more frequently detected in those with *BRAF*-altered CRC compared to those with *BRAF*-altered non-CRC GI malignancies (31% vs. 6.8%, *p* < 0.001). Similarly, MSI-H was more frequently detected in BRAF-altered CRC versus non-CRC GI malignancies (30% vs. 4%, *p* < 0.001) (Figure 3, Appendix A). Mismatch repair protein (MMR) status by immunohistochemistry was available for a subset of patients (*n* = 730). MMR deficiency was also noted more commonly in patients with CRC compared to non-CRC GI cancers (36% vs. 6.3%, *p* < 0.001). In patients with MSI-stable tumors, rates of TMB-high were similar in CRC and non-CRC GI cancers (2.1% vs. 2.7%, *p* = 0.5).

### 3.5. Co-Mutations in BRAF-Altered Tumors

As part of an exploratory analysis, the co-mutational landscape of *BRAF*-altered cancers was evaluated (Appendix A). Commonly occurring co-mutations seen in *BRAF*-altered CRC were *TP53* (70%), APC (42%), *RNF43* (30%), *SMAD4* (22%), *PIK3CA* (21%), *KMT2D* (detected in tissue only), and *MSH3* (18%). The most common co-mutations observed in non-CRC GI malignancies were *TP53* (52%), *CDKN2A* (32%), *CDKN2B* (20%, detected in tissue only), *ARID1A*, *SMAD4*, *KRAS* (17%), and *MTAP* (16%, detected in tissue only).

We specifically looked at the association of BRAF and RAS alteration. Of 2516 patients with a *BRAF* alteration, 379 (15%) patients had a co-alteration in an *RAS* gene. Interestingly, amongst 27 patients with *BRAF* amplification, 37% of the patients had an alteration in an RAS gene.

### 3.6. Characterization of Patients Treated with a BRAF Inhibitor

Last, we sought to characterize patients who had received a BRAF inhibitor. However, only 52 (2.8%) patients with *BRAF*-altered CRC and 11 (1.6%) patients with a *BRAF*-altered non-CRC GI malignancy had received a BRAF inhibitor, limiting further analysis. The most commonly used BRAF inhibitor among patients with *BRAF*-altered CRC was encorafenib in 85%, whereas the most common BRAF inhibitor in those with a *BRAF*-altered non-CRC GI malignancy was dabrafenib in 91%.

## 4. Discussion

In the present study, we described the frequency and spectrum of *BRAF* alterations in non-CRC GI malignancies by using a large database of patients who underwent tumor somatic profiling and compared it to the frequency of *BRAF* alteration in CRC. This has not been widely reported previously. *BRAF* alterations were present in 8.9% of CRC patients, yet less frequently detected in patients with a non-CRC GI malignancy (2.2%). In addition, non-CRC GI malignancies more frequently harbored uncommon class II and III *BRAF* alterations, including non-V600E SVs, fusions, and CNAs.

Our data illustrates that there is a unique profile of *BRAF* alterations in the non-CRC GI malignancy population. Among patients with CRC, V600E accounted for 75% of *BRAF* alterations, which is consistent with prior reports [5,15]. Conversely, V600E only accounted for 27% of *BRAF* mutations among non-CRC GI malignancies, and class II and class III alterations accounted for 28% and 23%, respectively. Of note, a previous pan-cancer analysis of 115,000 patients found that non-V600E *BRAF* mutations accounted for 35% of *BRAF* mutations, which is lower than the rates of class II and III mutations reported in this study [16]. The difference in frequency of non-V600E *BRAF* mutation between CRC and non-CRC GI malignancies is important to consider, recognizing variation in tumor behavior by class of *BRAF* mutation. In the case of CRC, class II *BRAF* mutations have shown similar clinical outcomes when compared to V600E*,* while class III *BRAF* mutations have been associated with relatively better prognosis [5]. How these different classes of mutation may impact outcomes in non-CRC GI malignancies is not well established at this time and merits further evaluation. A previous study evaluating clinical outcomes in patients with *BRAF*-mutated intrahepatic cholangiocarcinoma reported that V600E was associated with larger, more invasive tumors and poorer overall survival when compared to non-V600E mutations. However, the breakdown of these mutations with regard to different classes was not reported [17]. Prognostic implications of non-V600E mutation also merit further evaluation across non-CRC GI malignancies.

When looking at non-CRC GI malignancies, biliary tract cancers were found to have the highest frequency of *BRAF* alteration in 4.1% of patients, which is comparable to prior studies reporting a frequency of 5–7% [18]. *BRAF* V600E is a targetable mutation in biliary tract cancers, based on the results of the phase 2 basket trial, ROAR [19]. However, it is again important to note the high frequency of non-V600E mutations among non-CRC GI malignancies. In the case of small bowel cancer, which was the second most common non-CRC GI malignancy to have a *BRAF* mutation in this study, an incidence of 4.0% was detected, which is lower than a prior report suggesting a frequency of approximately 9% [20]. Studies have also shown that the majority of *BRAF* mutations in small bowel adenocarcinoma are non-V600E, which is consistent with our current report [21,22]. Overall, these two examples demonstrate an unmet need for targeting non-V600E *BRAF* alteration in non-CRC GI malignancies. There is some evidence that MEK inhibition may have activity in targeting non-V600E *BRAF* mutations [23,24,25]. However, an additional basket study including patients with solid tumors and lymphoma harboring a non-V600E *BRAF* mutation or *BRAF* fusion did not show promising activity [26]. Additionally, there is some thought that class III BRAF-mutated CRC may have elevated sensitivity to EGFR inhibition, which may be applicable to non-CRC GI malignancies as well [27,28].

Another finding of interest in this study was higher rates of *BRAF* fusions in patients with *BRAF*-altered non-CRC GI malignancies compared to CRC (12% vs. 2.2%, *p* < 0.001). *BRAF* fusions result in the substitution of the N-terminus region of *BRAF*, which is its inhibitor domain, with another binding partner, which may result in constitutive activation of the *BRAF* gene [29,30]. A prior study, including 2154 patients with CRC who completed comprehensive genomic profiling found *BRAF* fusion in 4 patients with CRC (0.3%) [31]. Concerning non-CRC GI malignancies, fusions were most frequently detected in pancreatic cancer, with an incidence of 0.3% in the aforementioned study [31]. In the current study, with regards to non-CRC GI tumors, we also found that fusions were more frequent in pancreatic as well as esophageal tumors, accounting for 25% and 10% of *BRAF* alterations occurring in those tumor types, respectively. While rare, solid malignancies with *BRAF* fusion may be targeted by MEK inhibitors, based on a study evaluating trametinib in melanoma with non-V600E *BRAF* mutation or *BRAF* fusion, although additional study in other solid malignancies is needed [23].

In addition, we found that *BRAF*-altered CRC had a significantly higher frequency of MSI-H and TMB-H when compared to *BRAF*-altered non-CRC GI malignancies. The association between *BRAF* mutation and MSI-H in CRC is well established. The reason for the decreased frequency of MSI-H among patients with *BRAF*-mutated non-CRC GI malignancies is not certain, but prior work has demonstrated that non-V600E *BRAF*-mutated CRC exhibits lower rates of MSI-H [32]. Recognizing that non-CRC GI malignancies have higher rates of non-V600E *BRAF* mutation, this is a very plausible explanation for this difference in frequency.

Limitations of this study include a lack of diversity among the sample population ordered at a large diagnostic testing laboratory, which was predominantly white, limiting generalizability to other racial groups to some extent. Additionally, data regarding *BRAF* inhibitor treatment and clinical outcomes has yet to be evaluated. In this study, only a small population of patients had a history of receiving *BRAF* inhibitors prior to sample collection (*n* = 63); thus, this remains an important area for future research in exploring outcomes of patients with various *BRAF* alterations across tumor subtypes to determine if *BRAF* inhibitors can alter the natural history of the disease. Importantly, it will be important to assess the impact of *BRAF* inhibitors in *BRAF* non-V600E mutated non-CRC GI cancers, especially those with *BRAF* fusions.

## 5. Conclusions

In this study, we report a higher frequency of non-V600E (class II and III) *BRAF* alterations in non-CRC GI malignancies compared to CRC. Prior work has shown that these mutations result in distinct phenotypes and may have prognostic implications for the patient. Future work should aim to better define the prognostic implications of these mutations, as well as the efficacy of targeted therapies to treat this subpopulation of cancer patients.

## Figures and Tables

**Figure 1 cancers-16-01823-f001:**
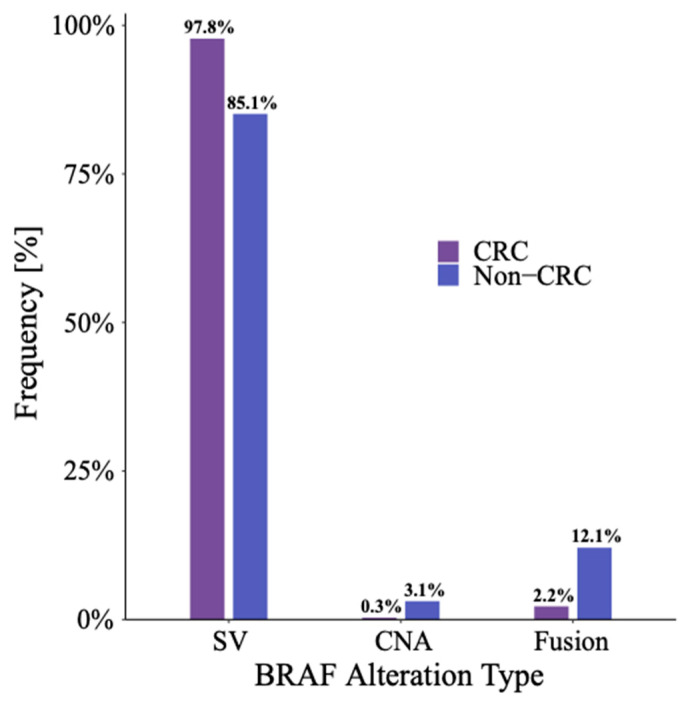
Frequency of *BRAF* gain-of-function alteration by primary cancer location.

**Figure 2 cancers-16-01823-f002:**
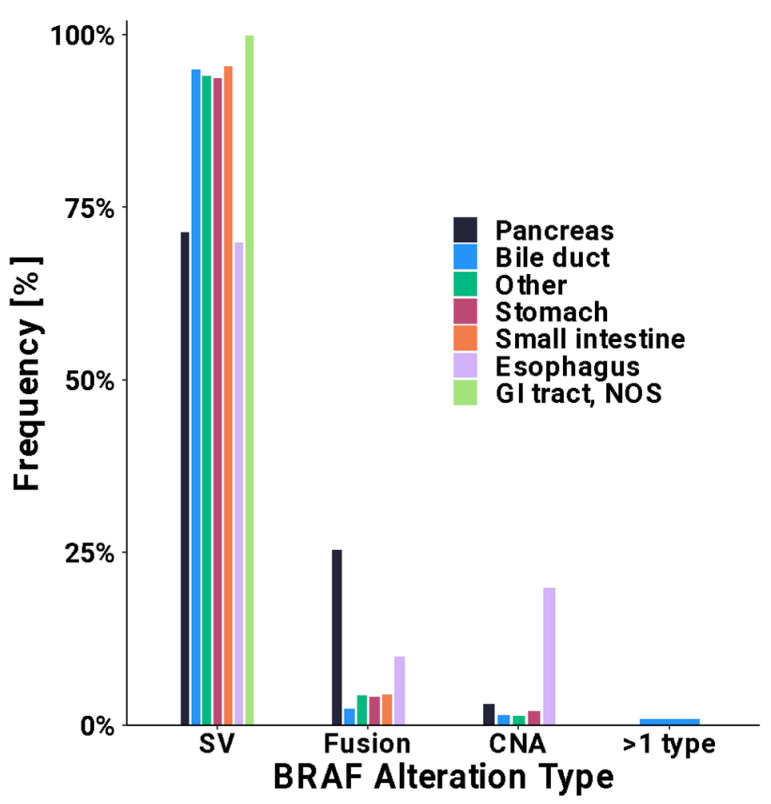
Frequency of *BRAF* gain-of-function alteration type by cancer site in patients with non-colorectal cancer gastrointestinal malignancies.

**Figure 3 cancers-16-01823-f003:**
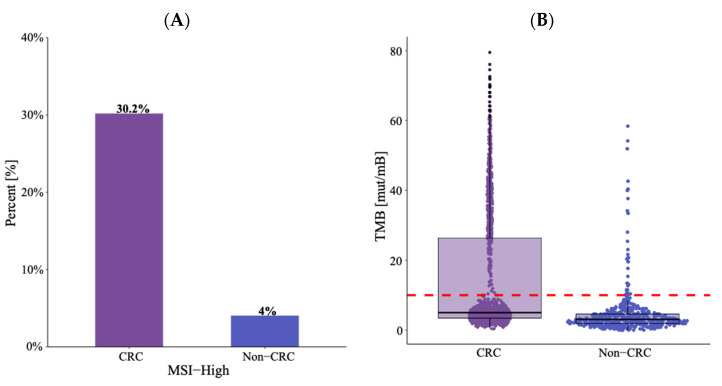
Tissue-based (**A**) microsatellite instability and (**B**) tumor mutation burden among patients with *BRAF* gain-of-function mutation by primary cancer location. Dashed line in figure B represents TMB cut off of 10 mutations/Mb.

**Table 1 cancers-16-01823-t001:** Demographics and clinical characteristics of patients with a *BRAF* gain-of-function alteration by primary cancer location.

	Overall, *n* = 2516 ^1^	CRC, *n* = 1838 ^1^	Other GI, *n* = 678 ^1^	*p*-Value ^2^
Age at diagnosis	67 (57, 75)	67 (57, 76)	66 (58, 73)	0.029
Unknown	31	23	8	
Age at diagnosis				0.13
>65	1349 (54%)	1002 (55%)	347 (52%)	
≤65	1136 (46%)	813 (45%)	323 (48%)	
Unknown	31	23	8	
Gender				<0.001
Female	1351 (54%)	1043 (57%)	308 (45%)	
Male	1165 (46%)	795 (43%)	370 (55%)	
Race				<0.001
White	1229 (83%)	924 (85%)	305 (77%)	
Black or African American	113 (7.6%)	69 (6.4%)	44 (11%)	
Other	90 (6.1%)	67 (6.2%)	23 (5.8%)	
Asian or Pacific Islander	49 (3.3%)	24 (2.2%)	25 (6.3%)	
Unknown	1035	754	281	
Ethnicity				0.7
Not Hispanic or Latino	759 (89%)	571 (89%)	188 (89%)	
Hispanic or Latino	91 (11%)	67 (11%)	24 (11%)	
Unknown	1666	1200	466	
Initiation of BRAF inhibitor prior to sample collection	63 (2.5%)	52 (2.8%)	11 (1.6%)	0.086
Primary cancer site				
Colon	1556 (62%)	1556 (85%)	0 (0%)	
Pancreas	259 (10%)	0 (0%)	259 (38%)	
Rectum	244 (9.7%)	244 (13%)	0 (0%)	
Intrahepatic bile duct	126 (5.0%)	0 (0%)	126 (19%)	
Stomach	48 (1.9%)	0 (0%)	48 (7.1%)	
Rectosigmoid junction	38 (1.5%)	38 (2.1%)	0 (0%)	
Esophagus	30 (1.2%)	0 (0%)	30 (4.4%)	
Biliary tract	29 (1.2%)	0 (0%)	29 (4.3%)	
Gastrointestinal tract	29 (1.2%)	0 (0%)	29 (4.3%)	
Extrahepatic bile duct	26 (1.0%)	0 (0%)	26 (3.8%)	
Ampulla of Vater	23 (0.9%)	0 (0%)	23 (3.4%)	
Duodenum	19 (0.8%)	0 (0%)	19 (2.8%)	
Gallbladder	19 (0.8%)	0 (0%)	19 (2.8%)	
Liver	19 (0.8%)	0 (0%)	19 (2.8%)	
Appendix	18 (0.7%)	0 (0%)	18 (2.7%)	
Small intestine	14 (0.6%)	0 (0%)	14 (2.1%)	
Jejunum	10 (0.4%)	0 (0%)	10 (1.5%)	
Anus	8 (0.3%)	0 (0%)	8 (1.2%)	
Ileum	1 (<0.1%)	0 (0%)	1 (0.1%)	
Assay				1.00
xT (tissue-based)	1833 (73%)	1339 (73%)	494 (73%)	
xF (liquid-based)	683 (27%)	499 (27%)	184 (27%)	
Months from diagnosis to sample collection				<0.001
Median (IQR)	1 (0, 9)	1 (0, 10)	0 (0, 5)	
Range	0, 225	0, 225	0, 173	
Stage within 60 days of sample collection				<0.001
Stage 4	1317 (78%)	970 (76%)	347 (83%)	
Stage 3	244 (14%)	208 (16%)	36 (8.6%)	
Stage 2	107 (6.3%)	84 (6.6%)	23 (5.5%)	
Stage 1	19 (1.1%)	7 (0.6%)	12 (2.9%)	
Unknown	829	569	260	
Metastases prior to sample collection	1660 (66%)	1258 (69%)	402 (59%)	<0.001
Unknown	11	9	2	

^1^ median (IQR); *n* (%) ^2^ Wilcoxon rank sum test; Pearson’s chi-squared test.

**Table 2 cancers-16-01823-t002:** Description of cohort and incidence of *BRAF* gain-of-function mutation by primary tumor location.

Characteristic	Overall, *n* = 51,560 ^1^	BRAF_wt, *n* = 49,044 ^2^	BRAF_GOF, *n* = 2516 ^2^	*p*-Value ^3^
Cohort				<0.001
Other GI	30,904 (60%)	30,226 (98%)	678 (2.2%)	
CRC	20,656 (40%)	18,818 (91%)	1838 (8.9%)	
Cancer group				<0.001
CRC	20,656 (40%)	18,818 (91%)	1838 (8.9%)	
Pancreas	14,300 (28%)	14,041 (98%)	259 (1.8%)	
Bile duct	4923 (9.5%)	4723 (96%)	200 (4.1%)	
Other	3434 (6.7%)	3366 (98%)	68 (2.0%)	
Esophagus	3316 (6.4%)	3286 (99%)	30 (0.9%)	
Stomach	3029 (5.9%)	2981 (98%)	48 (1.6%)	
Small intestine	1093 (2.1%)	1049 (96%)	44 (4.0%)	
GI tract, NOS	809 (1.6%)	780 (96%)	29 (3.6%)	

^1^ *n* (%), note that percentage indicates column percent. ^2^ *n* (%), note that percentage indicates row percent. ^3^ Pearson’s chi-squared test.

**Table 3 cancers-16-01823-t003:** *BRAF* gain-of-function mutation profile among all patients completing solid-tissue- or liquid-based assay (including specific alterations observed in ≥1% of patients with BRAF mutation).

	Overall, *n* = 2516 ^1^	CRC, *n* = 1838 ^1^	Other GI, *n* = 678 ^1^	*p*-Value ^2^	*q*-Value ^3^
BRAF mutation class					
Class I	1574 (63%)	1385 (75%)	189 (28%)	<0.001	<0.001
Class II	314 (12%)	125 (6.8%)	189 (28%)	<0.001	<0.001
Class III	414 (16%)	259 (14%)	155 (23%)	<0.001	<0.001
BRAF mutation					
Val600Glu	1561 (62%)	1381 (75%)	180 (27%)	<0.001	<0.001
Asp594Gly	152 (6.0%)	114 (6.2%)	38 (5.6%)	0.6	0.6
Asp594Asn	95 (3.8%)	51 (2.8%)	44 (6.5%)	<0.001	<0.001
Gly469Ala	56 (2.2%)	29 (1.6%)	27 (4.0%)	<0.001	<0.001
Asn486_Pro490del	52 (2.1%)	0 (0%)	52 (7.7%)	<0.001	<0.001
Asn581Ser	39 (1.6%)	21 (1.1%)	18 (2.7%)	0.006	0.009
Lys601Glu	37 (1.5%)	18 (1.0%)	19 (2.8%)	<0.001	0.001
BRAF-SND1	32 (1.3%)	3 (0.2%)	29 (4.3%)	<0.001	<0.001
Gly466Val	32 (1.3%)	20 (1.1%)	12 (1.8%)	0.2	0.2
CN amp	27 (1.1%)	6 (0.3%)	21 (3.1%)	<0.001	<0.001
Gly466Glu	25 (1.0%)	15 (0.8%)	10 (1.5%)	0.14	0.2

^1^ *n* (%), ^2^ Pearson’s chi-squared test, ^3^ false discovery rate correction for multiple testing.

## Data Availability

Deidentified data used in the research was collected in a real-world health care setting and is subject to controlled access for privacy and proprietary reasons. When possible, derived data supporting the findings of this study have been made available within the paper and its Appendix A.

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
