# Peer review of "Frequency of Common and Uncommon BRAF Alterations among Colorectal and Non-Colorectal Gastrointestinal Malignancies"

_cancers, 2024, doi:10.3390/cancers16101823_

Round 1
Reviewer 1 Report
Comments and Suggestions for Authors
The present study “Frequency of Common and Uncommon BRAF alterations among Colorectal and
Non-colorectal Gastrointestinal Malignancies” reports a comprehensive assessment of BRAF mutation in a large cohort of patients with GI malignancies.
51,560 patients were evaluated, 40% with a diagnosis of CRC and 60% with non-CRC GI malignancies. BRAF gain of function alterations were reported more frequently in CRC (8.9%) than non-CRC GI malignancies (2.2%). The study evaluates BRAF mutation frequencies in the full cohort.
The study is well written and clear.
- Please provide eligibility criteria
- Number of patients with multiple primitive tumors
- If available, how many patients with metastatic disease received targeted anti-BRAF agents
- If available, how many patients received a methylation test assessment
Please provide further information on MMR IHC analysis; how many patients have absence of MLH1/PMS2
Author Response
1) Please provide eligibility criteria
Response: The eligibility criteria for this cohort included patients with a primary diagnosis of a GI malignancy were identified from the Tempus database and filtered to those who received Tempus xT (solid tumor panel) or xF (liquid biopsy) testing prior to September 2023. This information is included in the Methods section under “cohort selection” subheading.
2) Number of patients with multiple primitive tumors
Response: All of the included patients had a primary diagnosis of a GI cancer. The concept of a primary cancer is defined as the cancer diagnosis for which the patient underwent NGS testing. In our cohort, 17% of the patients had another prior malignancy with most common being breast cancer (3.2%), prostate cancer (2.1%), hematologic malignancy (1.9%) and skin cancer (0.8%). The rates of multiple primary cancers was similar in two groups. This information is now included in the methods and Results section
3) If available, how many patients with metastatic disease received targeted anti-BRAF agents
Response: In this study only 63 patients with metastatic disease received anti-BRAF agents with 52 patients in CRC group and 11 patients in non-CRC GI cancer group. The clinical treatment data is somewhat limited.
4) If available, how many patients received a methylation test assessment
Response: Unfortunately, we do not have methylation assessment data for this cohort
5) Please provide further information on MMR IHC analysis; how many patients have absence of MLH1/PMS2
Response: In the BRAF-altered group, 20% of the patients had MSI-high cancers, 30% in the CRC group and 4% in the non-CRC GI cancer group. MSI assessment methodology is described in detail in the manuscript. Mismatch repair protein (MMR) status by immunohistochemistry was available for a subset of patients (n=730). MMR deficiency was also noted more commonly in patients with CRC compared to non-CRC GI cancers (36% vs 6.3%). We have included this information in the manuscript.
Because of short turnaround time of 10 days required for response, we are unable to provide granular data on specific MMR proteins at this time. However, we expect the rates of MLH1/PMS2 deficiency to be consistent with prior studies.
Reviewer 2 Report
Comments and Suggestions for Authors
Dear Author,
I have reviewed the article discussing BRAF alterations GI tumors. It is indeed a meaningful study with over 50,000 samples enrolled. However, there has been considerable research published on the distribution of BRAF gene mutations in solid tumors. This study did not provide me with attractive new information, particularly lacking analysis of the correlation between BRAF mutations and clinical efficacy or prognosis in different GI tumors. The content of the study appears somewhat limited.
I would appreciate it if the author could supplement the above-mentioned aspects in their work.
Author Response
Response: We appreciate the thoughtful reviewer’s comments and agree that many studies have looked into BRAF mutations in different cancers. However, to our knowledge, this is the first study to look at frequency and spectrum of mutations specifically in GI cancers. In particular, we hope that comparison of BRAF-altered CRC and non-CRC GI cancers provides valuable information. This is particularly important with tumor agnostic approval of BRAF targeted agents. In our cohort only 63 patients received BRAF targeted agents. Unfortunately, due to the nature of the data collection, we do not have clinical outcomes information and we could not provide that analysis. We agree that clinical outcome information in a large cohort will be extremely valuable and hope that we could have access to that data in future.
Reviewer 3 Report
Comments and Suggestions for Authors
This paper deals with an exceptionally large dataset, which is a strong advantage of this study. Distinct distribution of class I, II and III mutations in different tumor types is a valuable finding.
It is risky to consider the results of tumor and plasma (liquid biopsy) analysis in the same calculations. At least, the frequencies of mutations in these groups have to be compared.
I do not think it is appropriate to pool all non-colorectal cancers (CRC) in the same group. It is not logical to consider, say, esophageal and pancreatic cancers as same entity only on the basis that they are gastrointestinal but not CRC tumors.
The data on frequency of BRAF mutations are of limited value if they are not adjusted to the RAS gene status. Of course, an overall frequency of BRAF activation is relatively low in pancreatic cancers because the vast majority of these tumors have an activation of MAPK pathway via RAS mutation. Furthermore, class I mutations are almost always mutually exclusive with RAS activation, while the data are more complex for class II and class III mutations.
Please comment on the incidence of exon 11 and exon 15 mutations
Association with high tumor mutation burden (TMB) has limited value if it is not adjusted to microsatellite instability (MSI) status. MSI-positive and MSI-negative tumors with high TMB should be considered separately.
Please consider whether BRAF amplification is co-incident or mutually exclusive with RAS mutations.
The main numerical data have to be presented in the text: the number of cases of each cancer type, the frequency of each class of mutations (overall and in RAS-mutation-negative tumors), etc. Age- and gender-related distribution may be commented if feasible.
Introduction must comment on the classification of BRAF mutations.
Author Response
1) This paper deals with an exceptionally large dataset, which is a strong advantage of this study. Distinct distribution of class I, II and III mutations in different tumor types is a valuable finding.
Response: Thanks for the kind comments
2) It is risky to consider the results of tumor and plasma (liquid biopsy) analysis in the same calculations. At least, the frequencies of mutations in these groups have to be compared.
Response: The frequency of mutations with tissue based and liquid based assay were somewhat similar. The most common mutations in both groups were p.Val600Glu, p.Asp594Gly and p.Asp594Asn. If the reviewer feel that this information will add value to the manuscript we can include this information as another supplementary table.
3) I do not think it is appropriate to pool all non-colorectal cancers (CRC) in the same group. It is not logical to consider, say, esophageal and pancreatic cancers as same entity only on the basis that they are gastrointestinal but not CRC tumors.
Response: This is a great point that there are different tumor types in non-CRC GI cancers. However, because of relatively low prevalence of BRAF alteration in non-CRC GI cancers we had combined tge group for meaningful analyses. Apart from cholangiocarcinoma, the frequency of BRAF alterations is less than 2%. Table 1 does provide frequency of BRAF alteration in different tumor type.
4) The data on frequency of BRAF mutations are of limited value if they are not adjusted to the RAS gene status. Of course, an overall frequency of BRAF activation is relatively low in pancreatic cancers because the vast majority of these tumors have an activation of MAPK pathway via RAS mutation. Furthermore, class I mutations are almost always mutually exclusive with RAS activation, while the data are more complex for class II and class III mutations.
Response: Thanks for the input. Of 2,516 patients with a BRAF alteration, 379 (15%) had a co-alteration in a RAS gene. Interestingly, amongst 27 patients with BRAF amplification, 37% of the patients had alteration in RAS gene. This information is now included in the results section.
5) Please comment on the incidence of exon 11 and exon 15 mutations
Response: 6.4% of CRC patients had a short variant mutation in exon 11 compared to 15% in non-CRC. 90% of CRC patients had a short variant mutation in exon 15, compared to 58% in non-CRC. We have included this information in the results section of the manuscript.
6) Association with high tumor mutation burden (TMB) has limited value if it is not adjusted to microsatellite instability (MSI) status. MSI-positive and MSI-negative tumors with high TMB should be considered separately.
Response: Thanks for the feedback. In patients with MSI-stable tumor, TMB-high was present in 2.1% of the patients with CRC and 2.7% of the patients with non-CRC GI cancers. This information is now included in the manuscript.
7) Please consider whether BRAF amplification is co-incident or mutually exclusive with RAS mutations.
Response: Interestingly, amongst 27 patients with BRAF amplification, 37% of the patients had alteration in RAS gene. This information is now included in the results section.
8) The main numerical data have to be presented in the text: the number of cases of each cancer type, the frequency of each class of mutations (overall and in RAS-mutation-negative tumors), etc. Age- and gender-related distribution may be commented if feasible.
Response: We agree and have now included this information in the manuscript.
9) Introduction must comment on the classification of BRAF mutations.
Response: Thanks for the suggestion, this information is now included in the Introduction section of the manuscript.
Round 2
Reviewer 3 Report
Comments and Suggestions for Authors
I respect the opinion of the authors of the paper.